# Learning by Comparing: Boosting Multimodal Affective Computing through Ordinal Learning

## Abstract

Multimodal affective computing aims to integrate information from multiple modalities for the analysis of human affective states, opinion tendencies, behavior intentions, etc. Previous studies primarily focus on approximating predictions to annotated labels, often neglecting the ordinal nature of affective states. In this paper, we address this issue by exploring ordinal learning, and a Multimodal Ordinal Affective Computing (MOAC) framework is designed to enhance the understanding of the nature of affective concepts. Specifically, we propose coarse-grained label-level ordinal learning that prompts the model to *learn to compare* in the label space, encouraging higher predictive values for samples annotated with larger labels over those with smaller labels. Moreover, a regularization loss is proposed to prevent the output distributions from deviating significantly from the annotated label distributions. Fine-grained feature-level ordinal learning is then performed via the feature difference operation and the neutral embedding. The former compares samples in the feature space, calculating the difference between features of different samples to generate 'new' features for a more robust training. The latter seeks to reduce the difficulty of prediction by estimating the difference between the target multimodal representations and a neutral reference. We first demonstrate MOAC in multimodal sentiment analysis, which is a regression task that aligns well with the function of ordinal learning. Then we extend MOAC to classification tasks including multimodal humor detection and sarcasm detection to evaluate its generalizability. Experiments suggest that MOAC outperforms state-of-the-art methods.

## CCS Concepts

• **Computing methodologies → Learning to rank**; *Supervised learning by regression*; *Supervised learning by classification*; • **Information systems → Multimedia information systems**.

## Keywords

Multimodal Data; Multimodal Affective Computing; Ordinal Learning; Sentiment Analysis

**ACM Reference Format:**
Anonymous Author(s). 2025. Learning by Comparing: Boosting Multimodal Affective Computing through Ordinal Learning. In *Proceedings of the ACM Web Conference 2025 (WWW '25)*. ACM, New York, NY, USA, 14 pages. https://doi.org/XXXXXXX.XXXXXXX

## 1 Introduction

With the development of social media and Internet, abundant multimodal contents have been posted on the Internet in every single day, many of which are presented in video forms for individuals to convey their opinion inclinations and affective states. As a crucial and rapidly emerging field in multimodal machine learning [2, 29], Multimodal Affective Computing (MAC) helps uncover the intentions and opinions that lie beneath and has gained substantial research interest recently [20, 40]. MAC could contribute to transforming the Internet into a more emotionally intelligent and responsive environment. By integrating information from acoustic clues, spoken language, and visual expressions, it provides advanced analysis of users' sentiment polarities, opinion tendencies, behavior intentions, emotional states, etc [4], which helps enhance the experience of human-computer interactions. In this paper, we focus on three downstream tasks of MAC, including Multimodal Sentiment Analysis (MSA) [65], Multimodal Humor Detection (MHD) [14], and Multimodal Sarcasm Detection (MSD) [5].

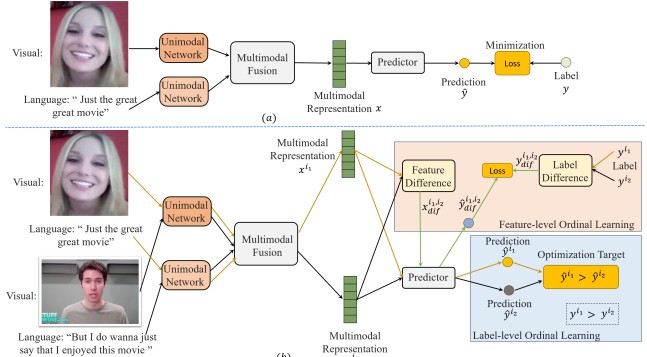

**Figure 1: A simple diagram of (a) traditional optimization strategy and (b) the proposed ordinal learning strategy. The neutral embedding is not shown in the figure for clarity.**

The majority of prior MAC methods focus on devising fusion strategies to thoroughly investigate the interactions among modalities, and subsequently make predictions based on the integrated multimodal representations [18, 33, 34, 50, 55, 59]. As shown in Fig. 1, traditional methods generally learn by conducting the main task of affective computing, which directly approximate predictions to annotated labels. However, it has been revealed that assigning absolute values to affective concepts is not only prone to noise but also unsuitable due to their subjective and ambiguous nature [47, 61]. Yannakakis et al. [61] argue that ordinal labels provide a more appropriate representation for affective states, suggesting that compared to assigning absolute values, the task of assigning reference-based (relative) values to subjective notions is better aligned with their underlying features. Humans also tend to perceive and interpret affective notions in an ordinal way. For example,

for sentiment perception, instead of exactly perceiving an absolute value of sentiment intensity, we typical understand humans' sentiment states via comparison, e.g., 'excellent' conveys more intense sentiment than 'good'. Leveraging the ordinal nature of affective states, we can provide the model with rich knowledge of affective comprehension and a better understanding of affective concepts.

Drawing inspiration from the way that humans perceive affective notions, we propose a Multimodal Ordinal Affective Computing (MOAC) framework to explore another perspective for affective understanding. MOAC learns by comparing affective concepts in a manner similar to human perception, which is realized by the design of label-level and feature-level ordinal learning. Specifically, in label-level ordinal learning, we require the model to *learn to compare in the label space*, encouraging samples with larger annotated labels to receive higher predicted values than those with smaller labels. By comparing the predictive values between multiple samples rather than directly predicting the absolute labels for individual samples, the model is encouraged to emulate human-like reasoning and uncover the semantic meanings of affective notions. Moreover, to prevent the model from deviating significantly from the distributions of the annotated labels, we further devise a regularization loss to align the distributions of the learned and the annotated label spaces, providing a better parameter initialization for the model.

Feature-level ordinal learning is then developed to perform fine-grained learning by *calculating the difference between features from different samples*. Specifically, we conduct feature difference operation based on two samples and then let the model to predict the label difference between them. In this way, the model is encouraged to learn features that reflect the label information (magnitudes and directions) in an explicit way. Moreover, this operation can generate abundant 'new' samples for a more robust training. In addition, we dynamically maintain a neutral embedding that enables ordinal learning during the prediction. The neutral embedding seeks to reduce the difficulty of prediction via inferring the polarities and intensities (directions and magnitudes) of labels in an easier way, which is achieved by estimating the difference between the target multimodal representations and a neutral reference.

We first demonstrate the proposed MOAC in the context of MSA, which is a regression task that predicts a sentiment value for each video utterance, naturally aligning well with our ordinal learning. Afterwards, we extend MOAC to classification tasks (including MHD and MSD) to verify the generalizability of MOAC.

To sum up, the contributions of MOAC are listed as below:

- We elaborately devise an ordinal learning framework to align with the nature of ordinal affective concepts. In this way, we encourage the model to think like humans, discover the semantic meanings of affective concepts, and understand the ordinal relationships between samples.
- We propose label-level ordinal learning that encourages the model to learn to compare, forcing the samples with larger labels to have higher predicted values than the samples with smaller labels. Moreover, a regularization loss is introduced to prevent the distributions of predictions from diverging significantly from the distributions of the annotated labels.
- Feature-level ordinal learning is devised to compare the features of two samples and generate the feature difference

for robust prediction. Moreover, a neutral embedding is introduced to perform ordinal learning during prediction and reduce the difficulty of prediction.
- MOAC outperforms state-of-the-art algorithms on multiple MAC tasks across various datasets.

## 2 Related Work

### 2.1 Multimodal Affective Computing

A plethora of previous research for MAC focus on devising fusion techniques to generate insightful multimodal representations [40, 48, 63, 67]. For instance, methods based on tensor fusion [24, 28, 62] are widely-used as they are capable of learning multimodal representations that possess substantial expressive capabilities. Furthermore, various methods utilize Kullback–Leibler divergence, canonical correlation analysis and information bottleneck to regulate the learning of unimodal distributions [9, 46, 49]. For example, the Information-Theoretic Hierarchical Perception (ITHP) framework [57] employs the information bottleneck principle to learn representations. ITHP identifies a primary modality and treats other modalities as detectors within the information pathway, which are utilized to distill the flow of information. In the wake of BERT's success [8], there has been a burgeoning trend towards fine-tuning large pre-trained transformer models using multimodal datasets [26, 44]. For example, Kim and Park [23] introduce multimodal masked language modeling and alignment prediction to further pre-train BERT, thereby effectively capturing intra- and inter-modal dynamics. Additionally, Liu et al. [30] employ advanced ensemble techniques to maximize the benefits of multiple pre-trained models (such as GPT-2 [41]) for both language and visual modalities. Recently, self-/weakly-supervised learning on multimodal datasets has garnered considerable interest [10, 32]. A prevalent approach is the application of contrastive learning to extract representations from multimodal data [6, 27, 39, 60]. For example, Hybrid Contrastive Learning (HyCon) [37] introduces a methodology for intra- and inter-modal contrastive learning, which promotes closer alignment of unimodal representations within the same category while encouraging greater separation between those of different categories.

Despite obtaining satisfactory results, these methods fail to consider the ordinal characteristics of affective concepts, merely calculating the loss between annotated labels and predictions for individual samples. In contrast, MOAC introduces feature- and label-level ordinal losses that capitalize on ordinal relationships between samples to bolster the model's comprehension of affective notions.

### 2.2 Ordinal Learning in Affective Computing

The annotation of affective concepts is inherently subjective and prone to ambiguity [17, 45, 47]. Yannakakis et al. [61] posit that ordinal labels provide a more appropriate way for representing affective notions, as assigning relative values to these subjective notions aligns more closely with their inherent characteristics than assigning absolute values. They introduce the preference learning paradigm to train affective models using ordinal data, and verify the advantages of relative annotation in affective computing. To address the challenge of sentiment annotation, Stoehr et al. [47] design a Bayesian generative model that learns a composite sentiment dictionary, incorporating an ordinal scale of ordered discrete

sentiment values within the learned dictionary. Contrary to these methods, MOAC seeks to harness the ordinal information inherently present in the annotations of affective concepts to enrich the model's knowledge of affective comprehension rather than attempting to resolve the annotation challenge.

To effectively utilize the ordinal information inherently hidden in emotional ranks, Han et al. [12] propose a deep learning framework based on COnsistent RAnk Logits (CORAL) for ordinal Speech Emotion Recognition (SER) tasks. This framework simplifies a multi-class ordinal SER task into a series of binary SER sub-tasks, each predicting whether an utterance's emotion exceeds a specific rank. Furthermore, Xie et al. [58] incorporate ordinal regression to establish an ordinal-aware sentiment space, utilizing a triplet loss to ensure that the feature distance between an anchor and its positive sample is less than that between the anchor and its negative sample. In contrast, our label-level ordinal loss operates directly on the label space rather than the feature space, aligning more closely with the optimization objectives. Additionally, we introduce feature difference operation and neutral embedding to thoroughly capture the ordinal relationships among samples and enable ordinal learning during prediction, which can learn more discriminative features. Finally, while Xie et al. [58] select a single positive and a single negative sample per anchor, our MOAC selects multiple sample pairs for training, providing richer information for optimization.

## 3 Algorithm

In this section, we describe our MOAC in detail, and the diagram of MOAC is presented in Fig. 2. MOAC is evaluated on the MSA, MHD, and MSD tasks. The input is an utterance [38], defined as a segment of a video bounded by a sentence. The videos are collected from the Internet or TV shows, showcasing the behavior intentions, opinion tendencies, and affective states of speakers. Typically, each utterance has three modalities, i.e., acoustic ($a$), visual ($v$), and language ($l$). For MHD and MSD, to better capture humor-related information, following prior methods [13, 36], Human Centric Feature (HCF) is additionally extracted from language to serve as the fourth modality (please see [13] for more details) and is denoted as $U_h \in \mathbb{R}^{T \times d_h}$ ($T$ is the sequence length). In particular, for MHD and MSD, each sample consists of a target punchline utterance and its preceding context utterance. We concatenate the punchline and context feature sequences in the time dimension to generate unimodal inputs $U_m \in \mathbb{R}^{T \times d_m}$ ($m \in \mathcal{M} = \{a, v, l, h\}$).

As the MSA task (which is a regression task that predicts continuous sentiment scores) aligns well with ordinal learning, we first take MSA as an example to illustrate the algorithm design, and then extend MOAC to classification tasks (MHD and MSD).

### 3.1 Model Pipeline

#### 3.1.1 Unimodal Learning Networks.
Generally, unimodal networks are utilized to extract unimodal representations $\{x_m \in \mathbb{R}^{T \times d} \mid m \in \{a, v, l\}\}$ based on the input features $\{U_m \in \mathbb{R}^{T \times d_m} \mid m \in \{a, v, l\}\}$ ($m \in \{a, v, l, h\}$ for MSD and MHD), where $d$ is the shared feature dimensionality. Due to space limitations, the detail introduction of unimodal networks is placed in Section A.1 of the Appendix.

#### 3.1.2 Fusion Network.
Given that the primary focus of MOAC is not the design of fusion strategies, we simply utilize a multi-layer

perception network to derive the multimodal representation:

$$x = \text{Fusion}(x_l \oplus x_a \oplus x_v; \theta_f) \in \mathbb{R}^{T \times d} \quad (1)$$

$$x \longleftarrow \frac{1}{T} \sum_{t=1}^{T} (x)_t \in \mathbb{R}^{1 \times d} \quad (2)$$

where $x \in \mathbb{R}^{1 \times d}$ is the multimodal representation. Notably, for MHD and MSD, $x_h$ is also used for fusion. Our experiments indicate that, even when employing a straightforward fusion network, MOAC achieves state-of-the-art performance, which further underscores the efficacy of the proposed MOAC. Due to space constraints, the architecture of the fusion network is delineated in the Appendix.

#### 3.1.3 Neutral Embedding.
After obtaining the multimodal representation $x$, the regular procedure is to feed $x$ into the final predictor to infer the prediction. Differently, in our framework, we innovatively introduce the neutral embedding in the prediction procedure, which enables the model to perform ordinal learning in the feature space by estimating the difference between each target multimodal representation and the neutral embedding. Based on the feature difference (i.e., the bias) that is input to the predictor, the model is able to infer the polarity and intensity of the sentiment in an easier way, reducing the difficulty of prediction.

The proposed neutral embedding is made up of two terms: a global embedding and a learnable bias embedding. The global embedding $x_g \in \mathbb{R}^{1 \times d}$ is initialized randomly before training and is subsequently updated using the following equation:

$$x_g \longleftarrow \lambda \times x_g + (1 - \lambda) \times \frac{1}{|\mathcal{S}_n|} \sum_{j \in \mathcal{S}_n} x_{neu}^j \quad (3)$$

where $\lambda$ is the hyperparameter, $x_{neu}^j$ denotes the representation of the $j^{th}$ neutral sample in the neutral set $\mathcal{S}_n$, and $|\mathcal{S}_n|$ is the number of neutral samples in $\mathcal{S}_n$ ($\mathcal{S}_n$ is selected from a batch of data $\mathcal{D}$).

However, the global embedding alone might not be a perfect choice to serve as the embedding that reveals the general properties and distributions of neutral samples. This is because in practice, we can only select a small proportion of neutral samples for each updating due to the memory limitation of hardware, which inevitably introduces noise. Moreover, the embeddings of neutral samples change across iterations, and it may be difficult to balance the contribution of the previous embedding and the current one. Therefore, we seek to learn a bias embedding $x_b \in \mathbb{R}^{1 \times d}$ that automatically compensates for the possible noise and error introduced by the global embedding. The final neutral embedding is thus defined as:

$$x_n = \frac{1}{2}(x_g + x_b) \quad (4)$$

Nevertheless, the introduction of bias embedding raises another question: How to train the bias embedding? We explore a simple and intuitive solution: We feed the neutral embedding $x_n$ to the predictor, and force the prediction to be zero. In this way, $x_n$ is encouraged to preserve the neutral sentiment nature, which ultimately modifies the bias embedding towards a suitable direction. The neutral loss is thus defined as:

$$\mathcal{L}_{neu} = ||\text{Predictor}(x_n; \theta_p)||^2 \quad (5)$$

where Predictor is the final predictor parameterized by $\theta_p$. Notably, here we stop the flow of gradients back to the neutral samples to avoid the possible interference with the regular training.

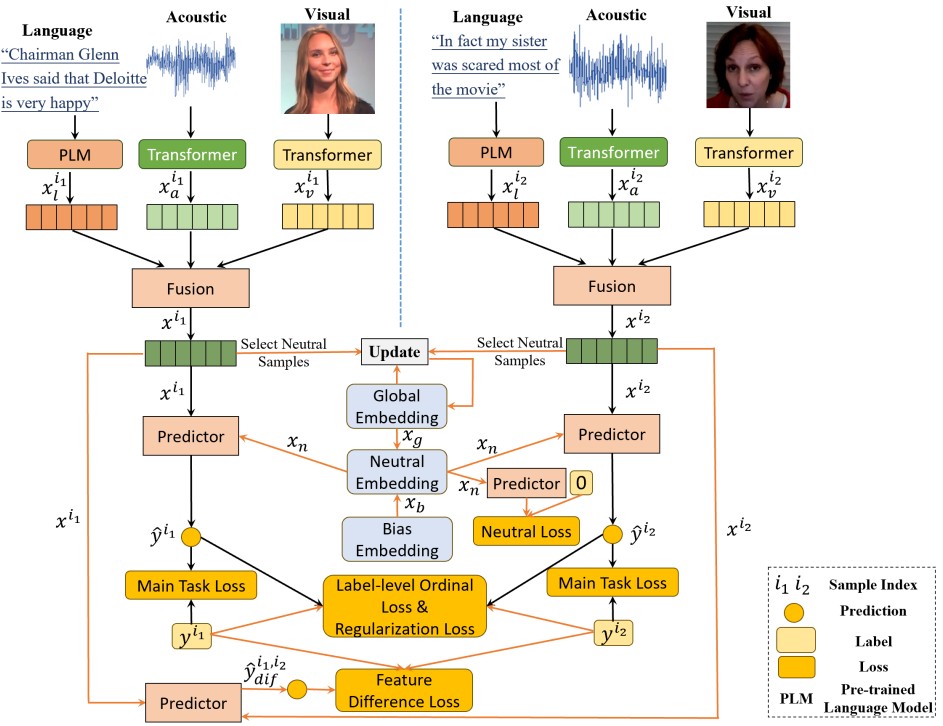

**Figure 2: Diagram of MOAC. Here we use two samples consisting of three modalities to illustrate the pipeline.**

*3.1.4* **Prediction Generation**. Finally, given the multimodal representation $x \in \mathbb{R}^{1 \times d}$ and neutral embedding $x_n \in \mathbb{R}^{1 \times d}$, the procedure for generating the prediction can be formulated as:

$$\hat{y} = \text{Predictor}(x - x_n; \ \theta_p) \in \mathbb{R}^1 \tag{6}$$

where $\hat{y}$ denotes the prediction. The predictive loss is defined as:

$$\mathcal{L}_p = \frac{1}{|\mathcal{D}|} \sum_{j \in \mathcal{D}} ||\hat{y}^j - y^j||^2 \tag{7}$$

where $y^j$ is the annotated label of sample $j$, $\mathcal{L}_p$ denotes the predictive loss, $\mathcal{D}$ is a collection of data (typically a mini-batch sampled from the whole dataset), and $|\mathcal{D}|$ is the number of samples in $\mathcal{D}$.

## 3.2 Ordinal Learning

*3.2.1* **Label-level Ordinal Learning**. Firstly, we conduct coarse-grained label learning via the proposed label-level ordinal learning. Leveraging the ordinal nature of affective notions to pre-train the model with rich knowledge of affective comprehension, this stage provides the model with a good initialization of parameters and a better understanding of affective states. Prior methods generally directly predict sentiment values to approximate the ground-truth labels [50, 57]. However, it has been revealed that assigning relative values instead of absolute values to affective notions is more suitable due to their subjective and ordinal nature [47, 61]. Inspired by this observation, we design the label-level ordinal loss to rank affective states of different samples rather than predicting absolute values for individual samples, encouraging the model to think like humans and grasp the intrinsic semantics of affective concepts.

Our ordinal loss is inspired by Bradley-Terry (BT) model [3] that aims to rank two samples and has been successfully applied in large language model training [42]. Specifically, the model preference

distribution between two samples can be written as:

$$
\begin{aligned}
p(\hat{y}^{i_1} > \hat{y}^{i_2}) &= \frac{exp(net(\{U_m^{i_1}\}; \theta))}{exp(net(\{U_m^{i_1}\}; \theta)) + exp(net(\{U_m^{i_2}\}; \theta))} \\
&= \frac{exp(\hat{y}^{i_1})}{exp(\hat{y}^{i_1}) + exp(\hat{y}^{i_2})}
\end{aligned} \tag{8}
$$

where $net$ denotes the whole model parameterized by $\theta$ (consisting of unimodal networks, fusion network and predictor), $\{U_m^{i_1}\}$ denotes the set of input features of sample $i_1$ for short (e.g., $\{U_m^{i_1}\} = \{U_a^{i_1}, U_v^{i_1}, U_l^{i_1}\}$ for MSA), $\hat{y}^{i_1}$ and $\hat{y}^{i_2}$ denote the predictive labels for sample $i_1$ and sample $i_2$, respectively. Notably, the label of sample $i_1$ should be larger than that of sample $i_2$. Provided a collection of data $\mathcal{D}$, we can conduct random sampling to select abundant comparison pairs for training the model via maximum likelihood. Simplifying Eq. 8 and take the logarithm of both sides of the equation, we have the following negative log-likelihood loss function:

$$p(\hat{y}^{i_1} > \hat{y}^{i_2}) = \frac{1}{1 + exp(\hat{y}^{i_2} - \hat{y}^{i_1})} = \sigma(\hat{y}^{i_1} - \hat{y}^{i_2}) \tag{9}$$

$$\mathcal{L}_{lo} = -\mathbb{E}_{(\{U_m^{i_1}\}, y^{i_1}) \in \mathcal{D}; \ (\{U_m^{i_2}\}, y^{i_2}) \in \mathcal{D}} [log \ \sigma(\hat{y}^{i_1} - \hat{y}^{i_2})] \tag{10}$$

where $\mathbb{E}$ denotes the expectation operation over all sampled pairs, $\sigma$ is the Sigmoid function, and $(\{U_m^{i_1}\}, y^{i_1}) \in \mathcal{D}$ denotes that sample $i_1$ comes from $\mathcal{D}$ (hereafter we use $i_1 \in \mathcal{D}$ for short). In this way, $\mathcal{L}_{lo}$ encourages the samples with more intense sentiment labels to have larger predictive values than those with less intense labels.

However, $\mathcal{L}_{lo}$ may encourage the output distributions to deviate from the distributions of the annotated labels, as it generally encourages the predictive values of the 'preferred'/'unpreferred' samples to become larger/smaller without imposing any constraints. Therefore, we add a regularization term to constrain the distributions

of the predictive labels to be close to those of the annotated labels. Specifically, we calculate the means and standard deviations of the predictive and annotated labels, and then align them with a margin:

$$\mathcal{L}_{re} = max(0, ||M_r - M_p||^2 - \gamma_1) + max(0, ||D_r - D_p||^2 - \gamma_1) \quad (11)$$

where $\mathcal{L}_{re}$ represents the regularization loss, $\gamma_1$ is the margin hyperparameter, $M_r$ and $M_p$ denote the averaged values for annotated labels and predictive labels respectively, and $D_r$ and $D_p$ denote the standard deviations for the annotated and predictive labels respectively. The existence of $\gamma_1$ allows the presence of a certain degree of distributional disparity, enabling the alignment to be easier (in the coarse-grained label learning stage, we do not require a fine-grained alignment between the learned and annotated labels). In this way, we force the predictive scores to have distributions similar to those of the annotated labels, providing a good initialization of the model parameters for the second stage.

Rather than predicting an exact sentiment score, the proposed ordinal loss forces the model to learn to compare, encouraging that samples with higher annotated values are assigned greater predictive values than those with lower annotated values. This optimization strategy is more similar to the human behavior of perceiving affective concepts and provides the model with richer insights into affective comprehension. In contrast to employing a triplet loss to perform ordinal learning in the feature space [58], our label-level ordinal loss conducts ordinal learning directly in the label space, aligning more closely with the optimization objective.

### 3.2.2 Feature-level Ordinal Learning.
The previous stage conducts coarse-grained label-level ordinal learning to help the model understand the semantic meanings and align with the ordinal nature of sentiment, providing a good initialization for parameters. In this stage, fine-grained ordinal learning is conducted to refine the model predictions and provide ordinal information at the feature-level.

Specifically, similar to label-level ordinal learning, given a collection of data $\mathcal{D}$, we conduct random sampling to select abundant comparison pairs $\mathcal{S}_c = \{(i_1, i_2)|i_1 \in \mathcal{D}, i_2 \in \mathcal{D}\}$ for training. Given a sample pair $(i_1, i_2) \in \mathcal{S}_c$, we first compute the difference between their multimodal representations and annotated labels:

$$y_{dif}^{i_1, i_2} = y^{i_1} - y^{i_2}, \quad x_{dif}^{i_1, i_2} = x^{i_1} - x^{i_2} \quad (12)$$

Notably, different from label-level ordinal learning, $y^{i_1}$ is not necessarily larger than $y^{i_2}$, which helps to avoid the predictive bias where the model tends to generate positive predictions. Then, we feed the feature difference $x_{dif}^{i_1, i_2}$ into the predictor to infer the prediction and compute the feature difference loss as:

$$\hat{y}_{dif}^{i_1, i_2} = \text{Predictor}(x_{dif}^{i_1, i_2}; \theta_p) \quad (13)$$

$$\mathcal{L}_{fd} = \mathbb{E}_{(i_1, i_2) \in \mathcal{S}_c}[max(0, ||\hat{y}_{dif}^{i_1, i_2} - y_{dif}^{i_1, i_2}||^2 - \gamma_2)] \quad (14)$$

Notably, here we do not require the model to exactly predict the label difference $y_{dif}^{i_1, i_2}$ based on the feature difference $x_{dif}^{i_1, i_2}$, but allow certain error via the introduction of the hyperparameter $\gamma_2$. This is because it is a very difficult task to explicitly predict the label difference between two different samples, which requires the model to have a deep understanding of the semantic meanings of the affective concepts. Moreover, the forced prediction of label difference may introduce noise to the model, because $y_{dif}^{i_1, i_2}$ is a computed

pseudo label that might be inaccurate under some circumstances. Since we use the same model pipeline to process the generated pseudo samples and the real samples, the noise might inevitably affect the learning of the network with respect to the real samples.

In addition to the feature difference operation, the incorporation of the neutral embedding can be seen as another form of feature-level ordinal learning (see Section 3.1.3). Therefore, the total loss of the feature-level ordinal learning is the sum of $\mathcal{L}_{fd}$ and $\mathcal{L}_{neu}$.

Compared to simply using triplet loss to conduct ordinal learning in the feature space via optimizing the feature distance between two samples [58], our feature difference operation directly predicts the label difference between two samples, using label information to learn discriminative features that reflect the polarities and intensities of the labels in an explicit way. In addition, the proposed operation can generates abundant 'new' samples, helping to train a more robust predictor. Moreover, we demonstrate that feature difference operation is similar to the widely-used mix-up operation [21, 54, 66]. Mix-up operation mixes the anchor with its augmented version or other sample from the same class, generating mixed features for training a robust model and thus addressing the limited training samples problem to some extent. Similar to mix-up operation, our method conducts learning based on the features of two samples to generate new features for training. Differently, our method computes the feature difference between two different samples, and the two samples are not necessarily from the same category, which can generate more diverse and enriched features. In Section 4.6, we demonstrate that feature difference operation slightly outperforms mix-up operation in our framework.

## 3.3 Model Optimization
The optimization of MOAC is divided into two stages: (1) Label-level ordinal learning stage: Coarse-grained label learning is performed in this stage to pre-train the model with abundant knowledge of affective comprehension and provides a good initialization for the parameters; (2) Feature-level ordinal learning stage: Fine-grained label learning is conducted in this stage via the main task loss (predictive loss, see Eq. 7) and the feature-level ordinal loss, equipping the model with fine-grained affective comprehension information.

Specifically, in the first stage, we optimize the model with the label-level ordinal loss and the regularization loss:

$$\mathcal{L} = \mathcal{L}_{lo} + \alpha \times \mathcal{L}_{re} \quad (15)$$

where $\alpha$ is the weight of the regularization loss. After obtaining a good initialization of model parameters, in the second stage, we train the model with feature-level ordinal loss and main task loss:

$$\mathcal{L} = \mathcal{L}_p + \beta \times (\mathcal{L}_{neu} + \mathcal{L}_{fd}) \quad (16)$$

where $\beta$ is the hyperparameter for the weight of feature-level ordinal loss. The auxiliary losses equip the model with extra knowledge of affective state comprehension, helping MOAC understand and interpret the semantic meanings of affective notions.

## 3.4 Extending to Classification Tasks
Our main task is MSA, which is a regression task that aligns well with the function of ordinal learning. Nevertheless, we demonstrate that MOAC can be extended to classification tasks. Due to space limitations, we place the analysis in Section A.2 of the Appendix.

 

## 4 Experiments

We conduct extensive experiments to evaluate MOAC using CMU-MOSI [65], CMU-MOSEI [64], UR-FUNNY [14], and MUStARD [5] datasets. Due to space limitations, the introduction of datasets, baselines, and experimental settings is placed in the Appendix. **Additional experimental results (e.g., hyperparameter robustness analysis, see Section A.8) are also shown in the Appendix.**

### 4.1 Results on MSA

We evaluate the performance of MOAC against competitive baselines using two well-established datasets for MSA. The results in Table 1 indicate that MOAC demonstrates superior results, consistently outperforming baselines across most of the evaluation metrics on both the CMU-MOSI and CMU-MOSEI datasets. Specifically, on CMU-MOSI, MOAC outperforms ITHP [57] that also uses DeBERTa [16] as the language network by 0.9% in Acc7, 0.5% in Acc2 and F1 score, and it outperforms UniMSE [19] that applies the powerful T5 [43] as the language network by over 2% and 2.5% in Acc2 and F1 score respectively. Furthermore, MOAC surpasses existing approaches in terms of MAE by a substantial margin. Notably, the improvement of MOAC is remarkable given that the state-of-the-art methods have already exceeded human performance [62]. On CMU-MOSEI, MOAC achieves improvements of 2.1%, 0.5%, and 0.5% over the state-of-the-art algorithm ITHP in terms of Acc7, Acc2, and F1 score, respectively. Additionally, MOAC outperforms UniMSE in terms of Acc2, F1 score, MAE, and Corr. Considering the results across two datasets, MOAC demonstrates state-of-the-art performance in MSA. In contrast to those methods that neglect the ordinal characteristics of affective concepts, MOAC introduces feature- and label-level ordinal learning to better align with the intrinsic ordinal nature of subjective notions and enrich the model's understanding of affective nuances. Notably, despite delivering commendable performance, MOAC has not yet achieved the highest scores in Acc7. We hypothesize that this shortfall may be partly attributed to the intentional introduction of the margin in the ordinal learning. The margin is designed to reduce the noise introduced by ordinal learning but permit certain levels of error in the regularization and feature difference losses, which does not specifically tailor a more fine-grained learning strategy for sentiment labels.

Furthermore, compared with TMSON [58] that also incorporates ordinal learning, MOAC demonstrates a significant performance advantage across both datasets, which underscores the effectiveness of our ordinal learning strategy. In contrast to utilizing a triplet loss [56] solely for ordinal learning within the feature space [58], our ordinal losses are operated on both the label and feature spaces. The proposed label-level ordinal loss is more closely aligned with the optimization objective, facilitating the model's direct acquisition of label information. Moreover, unlike TMSON that implicitly learns label information by optimizing the feature distance between a pair of samples, our feature difference loss explicitly predicts the label disparity between a pair of samples and leverages label information for direct feature optimization, enhancing the discriminative power of the learned features with respect to the labels.

These findings underscore the efficacy of our MOAC, highlighting the significance of conducting ordinal learning that is attuned to the intrinsic nature of affective concepts.

### 4.2 Results on MHD and MSD

To further evaluate the generalizability of MOAC with respect to multimodal classification tasks, we carry out experiments on the tasks of MHD and MSD using UR-FUNNY [14] and MUStARD [5] datasets. As presented in Table 2 and Table 3, for the MHD and MSD task, the proposed MOAC outperforms state-of-the-art methods HKT [13], MCL [35] and MGCL [36]. As for the trainable parameters, MOAC has fewer parameters than the baselines due to the use of simple fusion network and predictor. Therefore, the space complexity of the proposed MOAC is acceptable. Generally, MOAC reaches state-of-the-art performance with lower complexity in the MHD and MSD tasks, demonstrating the effectiveness and the generalizability of MOAC to the multimodal classification tasks.

### 4.3 Ablation and Comparison Experiments

In this section, we conduct extensive ablation experiments to evaluate the effectiveness of each component in MOAC:

**(1) The importance of Ordinal Learning**: In the case of 'W/O Ordinal Learning', we remove the ordinal learning such that our MOAC becomes a regular multimodal model. As shown in Table 4, the performance of the model drops by 5.5%, 3.8%, 3.8% in Acc7, Acc2 and F1 score respectively, demonstrating the importance of learning ordinal relationships between samples to uncover the semantic meanings and inherent nature of affective concepts;

**(2) Discussion on Label-level Ordinal Learning**: In 'W/O Label-level Ordinal Learning', we remove the label-level ordinal loss and the accompanied regularization loss. The results suggest that the model's performance declines by approximately 1.5% in Acc7, Acc2 and F1 score, revealing that it is of great significance to learn ordinal relationships between samples in the label space, which helps understand the nature of affective concepts. Additionally, solely removing the regularization loss leads to a decrease in performance by 2.3%, 0.9%, 0.9% in Acc7, Acc2 and F1 score, respectively (see 'W/O Regularization Loss'), demonstrating the effectiveness of regularization loss for regularizing the output distributions and providing a better initialization for the parameters;

**(3) Discussion on Feature-level Ordinal Learning**: In the case of 'W/O Feature-level Ordinal Learning', we remove neutral embedding, neutral loss, and feature difference operation. The performance of MOAC exhibits a significant decline, and the decline is greater than when label-level ordinal learning is removed. This is reasonable because feature-level ordinal learning can reduce the difficulty of prediction, learn more discriminative features that reflect the magnitudes and directions of sentiment labels in an explicit way, and generate abundant 'new' features to enhance model robustness. We also investigate the importance of different operations in feature-level ordinal learning. As we can infer from the results of 'W/O Feature Difference Operation' and 'W/O Neutral Embedding', these two techniques both contribute significantly to the improvement of the model. We argue that this is because the feature difference operation encourages the model to directly learn features that reflect label information and generates abundant features for a more robust training, and neutral embedding helps the model infer the polarities and intensities of labels in an easier way via estimating the difference between the target multimodal representations and a neutral reference. Additionally, we estimate

**Table 1: The results on CMU-MOSI and CMU-MOSEI. Apart from TMSON, ConFEDE and UniMSE, the results of baselines are derived from our experiments. The best results are highlighted in bold, and the runner-up results are indicated with underlines.**

| | CMU-MOSI | | | | | CMU-MOSEI | | | | |
|---|---|---|---|---|---|---|---|---|---|---|
| | Acc7↑ | Acc2↑ | F1↑ | MAE↓ | Corr↑ | Acc7↑ | Acc2↑ | F1↑ | MAE↓ | Corr↑ |
| Graph-MFN [64] | 34.4 | 80.2 | 80.1 | 0.939 | 0.656 | 51.9 | 84.0 | 83.8 | 0.569 | 0.725 |
| MFM [51] | 33.3 | 80.0 | 80.1 | 0.948 | 0.664 | 50.8 | 83.4 | 83.4 | 0.580 | 0.722 |
| MMIM [11] | 45.0 | 85.1 | 85.0 | 0.738 | 0.781 | 53.1 | 85.1 | 85.0 | 0.547 | 0.752 |
| HyCon [37] | 46.6 | 85.2 | 85.1 | 0.741 | 0.779 | 52.8 | 85.4 | 85.6 | 0.554 | 0.751 |
| UniMSE† [19] | 48.7 | 86.9 | 86.4 | 0.691 | 0.809 | 54.4 | 87.5 | 87.5 | 0.523 | 0.773 |
| ConFEDE† [60] | 42.3 | 85.5 | 85.5 | 0.742 | 0.782 | 54.9 | 85.8 | 85.8 | 0.522 | 0.780 |
| TMSON† [58] | 47.4 | 87.2 | 87.2 | 0.687 | 0.809 | 55.6 | 86.4 | 86.2 | 0.526 | 0.766 |
| MGCL [36] | 49.3 | 86.7 | 86.7 | 0.685 | 0.707 | 53.9 | 86.4 | 86.4 | 0.535 | 0.772 |
| ITHP [57] | 47.7 | 88.5 | 88.5 | 0.663 | 0.856 | 52.2 | 87.1 | 87.1 | 0.550 | 0.792 |
| MOAC | 48.6 | **89.0** | **89.0** | **0.605** | **0.857** | 54.3 | **87.6** | **87.6** | **0.512** | **0.793** |

**Table 2: The comparison with baselines on UR-FUNNY. The results of the baselines labeled by † are taken from [13], and other results are obtained in our own experiments.**

| Model | Accuracy | Number of Parameters |
|---|---|---|
| MISA-ALBERT† [15] | 69.82 | - |
| MAG-ALBERT† [44] | 72.43 | - |
| HKT† [13] | 77.36 | - |
| HKT [13] | 76.46 | 17,066,564 |
| MCL [35] | 77.67 | 13,762,973 |
| MGCL [36] | 78.06 | 14,062,342 |
| MOAC | **78.57** | **13,272,272** |

**Table 3: The comparison with baselines on MUStARD.**

| Model | Accuracy | Number of Parameters |
|---|---|---|
| MISA-ALBERT† [15] | 66.18 | - |
| MAG-ALBERT† [44] | 69.12 | - |
| HKT† [13] | 79.41 | - |
| HKT [13] | 76.47 | 17,101,372 |
| MCL [35] | 77.94 | 13,828,449 |
| MGCL [36] | 77.94 | 14,282,000 |
| MOAC | **80.88** | **13,454,832** |

**Table 4: Ablation experiments on CMU-MOSI dataset.**

| | Acc7 | Acc2 | F1 | MAE | Corr |
|---|---|---|---|---|---|
| W/O Ordinal Learning | 43.1 | 85.2 | 85.2 | 0.690 | 0.817 |
| W/O Label-level Ordinal Learning | 47.0 | 87.5 | 87.5 | 0.639 | 0.843 |
| W/O Regularization Loss | 46.3 | 88.1 | 88.1 | 0.630 | 0.847 |
| W/O Feature-level Ordinal Learning | 46.9 | 87.0 | 87.1 | 0.639 | 0.848 |
| W/O Feature Difference Operation | **48.9** | 87.9 | 88.0 | 0.624 | 0.848 |
| W/O Neutral Embedding | 46.9 | 88.1 | 88.1 | 0.626 | 0.850 |
| W/O Margin $\gamma_2$ | 48.3 | 88.7 | 88.7 | 0.607 | 0.860 |
| W/O Global Embedding | 48.8 | 88.7 | 88.7 | 0.606 | **0.862** |
| W/O Bias Embedding | 47.9 | 88.5 | 88.5 | 0.624 | 0.850 |
| MOAC | 48.6 | **89.0** | **89.0** | **0.605** | 0.857 |

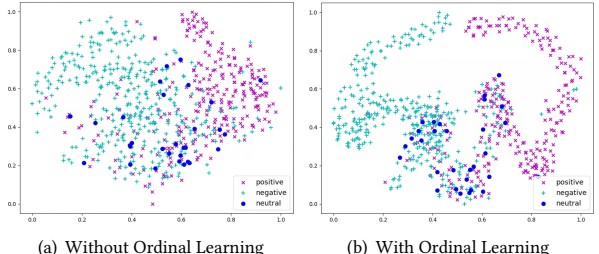

(a) Without Ordinal Learning          (b) With Ordinal Learning

**Figure 3: T-SNE visualization of multimodal features.**

the effectiveness of the margin $\gamma_2$ in feature difference loss, and the results of 'W/O Margin $\gamma_2$' suggest that $\gamma_2$ slightly improves the performance of the model, which can reduce the noise introduced by the generated features of the feature difference operation.

**(4) Further Discussion on Neutral Embedding**: Moreover, we estimate the roles of global embedding and learnable bias embedding in the neutral embedding, respectively. As we can infer from the results of 'W/O Global Embedding' and 'W/O Bias Embedding', when the global/bias embedding is removed, the model performance exhibits a slight decline, but it still obtains satisfactory results, indicating the effectiveness of bias/global embedding. Furthermore, the learnable bias embedding is more effective than the global embedding, verifying our assumption that adding a learnable embedding to the global embedding can reduce the noise and better represent the general properties of the neutral samples.

### 4.4 Visualization of Multimodal Features

We provide the visualization for the distributions of multimodal representations using the t-SNE [52] algorithm on the CMU-MOSI

testing set, and the results are illustrated in Figure 3. We can observe that compared to the situation that the ordinal learning is removed, when the ordinal learning is applied, the positive and negative data points are more distinctly separated, with data points within the positive/negative category being more tightly and centrally clustered. Additionally, the neutral data points are more concentrated in the middle between the positive and negative clusters. This indicates that the ordinal learning can help the model discover the relative sentiment magnitudes between different samples, possess a deeper understanding of the nature of sentiment, and learn more discriminative multimodal representations. Nevertheless, Figure 3 indicates that, although the neutral data points are situated between the positive and negative groups, there remains a degree of overlap and ambiguity with the positive and negative samples, which are not clearly differentiated. This issue might stem from the limited number of neutral utterances, hindering the model's ability to efficiently distinguish the neutral samples from the weakly positive



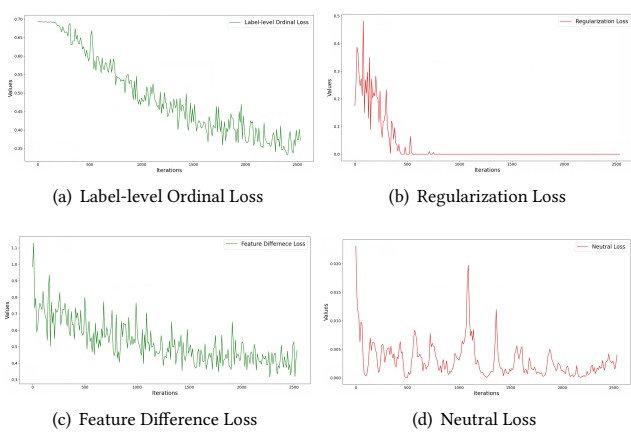

(a) Label-level Ordinal Loss

(b) Regularization Loss

(c) Feature Difference Loss

(d) Neutral Loss

**Figure 4: Learning curves of ordinal losses on CMU-MOSEI.**

**Table 5: Discussion of ordinal losses.**

|  | Acc7 (↑) | Acc2 (↑) | F1 (↑) | MAE (↓) | Corr (↑) |
|---|---|---|---|---|---|
| Reducing Sampled Pairs | 46.9 | 88.4 | 88.4 | 0.629 | 0.849 |
| Mix-up Operation | **49.3** | 88.9 | 88.9 | 0.612 | 0.852 |
| Ordinal Loss [58] | 48.9 | 88.7 | 88.7 | 0.614 | **0.857** |
| MOAC | 48.6 | **89.0** | **89.0** | **0.605** | **0.857** |

and weakly negative samples. This observation has inspired our future research to develop more effective representations for neutral utterances under conditions of insufficient data.

## 4.5 Learning Curves of Ordinal Losses

In this section, we provide the learning curves of ordinal losses on CMU-MOSEI dataset (note that here we average the results of every ten iterations to make the losses smoother). Firstly, for first-stage training, it can be seen from Fig. 4 (a) that the decline in the label-level ordinal loss is substantial, and the loss continues to drop sharply before the end of the first stage, indicating that the model can learn to capture information about the relative magnitudes of labels across different samples. For the regularization loss, at the early-stage of the first-stage training, the value of the regularization loss becomes zero, suggesting that it effectively aligns the distributions of the predictive and annotated labels. To our surprise, even though we do not conduct main task training in the first stage, the model still reaches satisfactory performance merely using the label-level ordinal learning (40.4% Acc7 and 86.0% Acc2 on CMU-MOSEI, 39.0% Acc7 and 87.3% Acc2 on CMU-MOSI), which further demonstrates the effectiveness of the proposed ordinal learning. The results on Acc7 also reveal that the label-level ordinal learning is coarse-grained, and that is why we need feature-level ordinal learning and main task learning to refine the predictions.

For the feature difference loss, it can be seen that although the loss has decreased, its rate of decline is not as significant as other losses, indicating that the feature difference operation is extremely challenging. As for the neutral loss, its value has already become very small in the initial phase of the second-stage training, which is reasonable because the global embedding has already to some extent integrated information from neutral samples and we only require the loss of one neutral embedding to become zero (i.e., the task is relatively easy via training the bias embedding).

## 4.6 Further Discussion on Ordinal Losses

Here we replace our ordinal losses with other losses to further evaluate their performance. Firstly, we design a simple variant for the feature difference loss and label-level ordinal loss, where the number of sampled pairs are the same as the batch size $|\mathcal{D}|$ (the default setting is twice the batch size). As shown in the 'Reducing Sampled Pairs' case in Table 5, the model's performance drops by over 1.5% in Acc7 and 0.5% in Acc2, and the performance on other metrics also decreases. This is reasonable because sampling limited training pairs is not sufficient for learning ordinal relationships between samples and cannot provide abundant ordinal information for optimization. Our ordinal losses sample abundant pairs to provide the model with richer knowledge about affective comprehension, which is shown to be effective in contrastive learning [22, 37].

We demonstrate that our feature difference operation is similar to mix-up operation [54, 66] in Section 3.2.2, and in this section we empirically analyze the advantage of feature difference operation over mix-up operation. Specifically, we replace feature difference operation with mix-up operation that mixes the features of two samples from the same category with equal weighting (the number of sampled pairs is the same as ours), and the results of 'Mix-up Operation' suggest that feature difference operation suppresses mix-up operation on all the metrics except for the Acc7. Combining the results on all the evaluation metrics, the feature difference operation slightly outperforms mix-up operation. We speculate that it is partly because our feature difference operation does not require that the selected two samples come from the same category, which can generate a more diverse and enriched set of features.

Furthermore, we replace the feature difference operation with the ordinal loss proposed in TMSON [58], which conducts ordinal learning in the feature space using a triplet loss [56]. The ordinal loss in TMSON uses hard sampling strategy, i.e., for each anchor $a$, the selected two samples for comparison (denoted as $r$ and $h$) satisfy the following constraint: $0 < |y^a - y^h| - |y^a - y^r| < \gamma$ where $\gamma$ is the defined minimum value. The ordinal loss is formulated as: $\mathcal{L}_{ord} = \frac{1}{n} \sum_{a=1}^{n} max(0, d(\mathbf{x}^a, \mathbf{x}^r) + \gamma_1 - d(\mathbf{x}^a, \mathbf{x}^h))$ where $d(*, *)$ is the distance function, $\gamma_1$ is the defined margin, and $n$ is the batch size. As shown in Table 5, when the feature difference loss is replaced with ordinal loss, the performance of the model declines in the majority of the evaluation metrics. We argue that this is because instead of optimizing the feature distance between two samples which learns the label information implicitly, the proposed feature difference operation predicts the label difference between two samples and directly optimizes the features using the label information, enabling the learned features to have higher discriminative power with respect to the labels.

## 5 Conclusion

We incorporate ordinal learning into MAC via the label- and feature-level ordinal learning. The label-level ordinal learning encourages the model to learn to compare in the label space. The feature-level ordinal learning computes the difference of the features from two samples to generate 'new' features. Particularly, we introduce neutral embedding to conduct ordinal learning during prediction and reduce the difficulty of prediction. Experiments indicate that MOAC reaches state-of-the-art performance on multiple MAC tasks.

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

# A Appendix

## A.1 Unimodal Learning Networks

In this section, we illustrate the structures of unimodal networks and the procedures to obtain the unimodal representations for the later fusion. Following the state-of-the-art models [13, 57], pre-trained language models [16, 25] are used to extract high-level language representations. Specifically, the procedures of the language learning network are shown as below:

$$
\begin{aligned}
\hat{X}_l &= \mathrm{PLM}(U_l; \; \theta_l) \in \mathbb{R}^{T \times d_l} \\
x_l &= (\hat{X}_l W_{pro} + b_{pro}) \in \mathbb{R}^{T \times d}
\end{aligned}
\tag{17}
$$

where PLM denotes the pre-trained language model, $U_l$ is the input token sequence and $T$ is the sequence length. $W_{pro} \in \mathbb{R}^{d_l \times d}$ and $b_{pro} \in \mathbb{R}^{1 \times d}$ are trainable parameters that map the output dimensionality of language learning network to the shared feature dimensionality $d$. For the acoustic and visual learning networks that are composed of transformer encoder layers [53], the pipelines are presented as follows:

$$
\begin{aligned}
\hat{X}_m &= \mathrm{Conv\,1D}(U_m; \; K_m) \in \mathbb{R}^{T \times d}, \; m \in \{a, v\} \\
x_m &= \mathrm{Transformer}(\hat{X}_m; \; \theta_m) \in \mathbb{R}^{T \times d}
\end{aligned}
\tag{18}
$$

where Conv 1D represents the temporal convolution whose kernel size $K_m$ is set to 3.

Please note that for MHD and MSD, the unimodal learning network of the HCF modality also consists of transformer encoder layers, which basically has the same structure as that of visual and acoustic modalities. Specifically, for MHD and MSD, the procedures of the transformer unimodal networks are illustrated as follows:

$$
\begin{aligned}
\hat{X}_m &= \mathrm{Transformer}(U_m; \; \theta_m) \in \mathbb{R}^{T \times d_m}, \; m \in \{a, v, h\} \\
x_m &= \mathrm{Conv\,1D}(\hat{X}_m; \; K_m) \in \mathbb{R}^{T \times d}
\end{aligned}
\tag{19}
$$

The generated unimodal representation $x_m$ is used for multimodal fusion.

## A.2 Extending to Classification Tasks

The main downstream task focused on this paper is MSA, which is a regression task that aligns well with the function of ordinal learning. In this section, we demonstrate that MOAC can be easily extended to multimodal classification tasks.

As any multi-class classification problem can be formulated as multiple binary classification tasks, we take binary classification problem as an example to illustrate the pipeline. For instance, for the MHD task, we need to predict whether the speaker expresses a humorous intention, where the model is required to output a probability $\sigma(\hat{y})$ with $\sigma$ being the Sigmoid function. The larger the value of $\sigma(\hat{y})$, the higher the likelihood that the speaker expresses a humorous intention. This aligns well with the nature of ordinal learning, and thus it is suitable to apply ordinal learning to learn humorous labels.

To be more specific, for the label-level ordinal learning, the selected comparison pairs shall come from different classes, and the loss function is still defined as:

$$
\mathcal{L}_{lo} = -\mathbb{E}_{(\{U_m^{i_1}\}, y^{i_1}) \in \mathcal{D}; \; (\{U_m^{i_2}\}, y^{i_2}) \in \mathcal{D}} \left[ \log \sigma(\hat{y}^{i_1} - \hat{y}^{i_2}) \right]
\tag{20}
$$

where $(\{U_m^{i_1}\}, y^{i_1})$ comes from the humorous class (1), and $(\{U_m^{i_2}\}, y^{i_2})$ comes from the non-humorous class (0). This is reasonable because this optimization objective encourages $\hat{y}^{i_2}/\hat{y}^{i_1}$ to be smaller/larger so that $\sigma(\hat{y}^{i_2})/\sigma(\hat{y}^{i_1})$ can approximate 0/1, which aligns well with the objective of binary classification tasks. As for the regularization term, since the means and standard deviations of the predicted and annotated labels are continuous values instead of discrete labels, the original form of the regularization loss shown in Eq. 11 is still applicable.

For feature-level ordinal learning, the neutral samples are the samples in which the speaker does not express humorous intention. Thus, the global embedding $x_g$ is updated using the non-humorous samples. Naturally, the calculation of $\mathcal{L}_{neu}$ in Eq. 5 becomes:

$$\mathcal{L}_{neu} = -log(1 - \sigma(\text{Predictor}(x_n; \theta_p))) \quad (21)$$

As for the feature difference operation, different from the original setting, in all the sampled pairs $\mathcal{S}_c = \{(i_1, i_2)|i_1 \in \mathcal{D}, i_2 \in \mathcal{D}\}$, the annotated label of sample $i_1$ should be larger than or equal to the label of sample $i_2$. If the annotated label of sample $i_1$ is larger than that of sample $i_2$ which means that $y^{i_1} = 1$ and $y^{i_2} = 0$, the label difference is equal to 1 (i.e., humorous). If the labels of sample $i_1$ and sample $i_2$ are the same, the label difference is equal to 0 (i.e., non-humorous). Therefore, the feature-level ordinal learning will not introduce bias into the predictions of the model. Finally, the feature difference loss is formulated as:

$$\mathcal{L}_{fd} = \mathbb{E}_{(i_1, i_2) \in \mathcal{S}_c} [max(0, -y_{dif}^{i_1,i_2} log\sigma(\hat{y}_{dif}^{i_1,i_2}) - (1 - y_{dif}^{i_1,i_2}) \log(1 - \sigma(\hat{y}_{dif}^{i_1,i_2})) - \gamma_2)] \quad (22)$$

## A.3 Datasets

We use the following datasets to evaluate the performance of the proposed MOAC:

(1) **CMU-MOSI** [65]: The CMU-MOSI dataset, widely utilized in Multimodal Sentiment Analysis (MSA), comprises over 2,000 video utterances collected from online websites. Each video utterance in this dataset is scored for sentiment intensity on a Likert scale from -3 to 3, where 3 represents the strongest positive sentiment and -3 represents the strongest negative sentiment.

(2) **CMU-MOSEI** [64]: CMU-MOSEI is a large scale multimodal dataset for MSA. It consists of more than 22,000 video utterances from over 1,000 YouTube speakers across 250 distinct topics. These utterances are randomly selected from a diverse array of topics and monologue videos. Each utterance is annotated from two perspectives: emotions categorized into six distinct values, and sentiment scores ranging from -3 to 3. We utilize the sentiment labels from the CMU-MOSEI dataset for our MSA task, which have the same sentiment scale as that of the CMU-MOSI dataset.

(3) **UR-FUNNY** [14]: The UR-FUNNY dataset is derived from TED talk videos with 1,741 speakers for the Multimodal Humor Detection (MHD) task. Each target video utterance in the UR-FUNNY dataset is called punchline, which encompasses language, acoustic, and visual modalities. The utterances preceding the punchline are the context utterances, which are fed into the model with the punchline for contextual analysis. The punchlines are identified using the 'laughter' tag in the transcripts, which indicates when the audience laughed during the talk. Negative samples are similarly identified, where the target punchline utterances are not followed by the 'laughter' tag. The UR-FUNNY dataset is divided into a training set with 7,614 instances, a validation set with 980 instances, and a testing set with 994 instances. In line with state-of-the-art methodologies [13, 35, 36], we employ version 2 of the UR-FUNNY dataset for our experimental analysis.

(4) **MUStARD** [5]: The MUStARD dataset is a Multimodal Sarcasm Detection (MSD) dataset sourced from popular television series such as Friends, The Big Bang Theory, The Golden Girls, and Sarcasmaholics. It comprises 690 video utterances that have been manually labeled as either sarcastic or non-sarcastic. The dataset includes the target punchline utterances along with the relevant preceding dialogues to provide contextual information.

## A.4 Evaluation Metrics

We assess the performance of the model on the MSA task using the following evaluation metrics: (1) Acc7: the accuracy of classifying sentiment scores into seven discrete classes; (2) Acc2: the binary accuracy for differentiating between positive and negative sentiments; (3) F1 score: a harmonic mean that balances precision and recall for binary sentiment classification; (4) MAE: the mean absolute error between the model's predictions and the annotated sentiment labels; and (5) Corr: the correlation coefficient indicating the strength and direction of the relationship between the predictions of the model and the human annotations. For Acc7, the model's predictions are rounded to the nearest integer within the scale from -3 to 3. When calculating Acc2 and F1 score, neutral utterances are not considered. And the neutral utterances are included in the calculations of MAE, Corr, and Acc7.

For MHD and MSD tasks, in alignment with prior methodologies [13, 35, 36], we report the binary accuracy (i.e., humorous or non-humorous, sarcastic or non-sarcastic) of the model.

## A.5 Feature Extraction Details

**Visual Modality**: Facet [1] is utilized to extract a series of visual features, including facial action units, facial landmarks, head pose, and others. These visual features are captured from each utterance, resulting in a temporal sequence that represents the facial expressions over time. For MHD and MSD, to be consistent with previous methods [13, 35], OpenFace 2 [1] is applied to extract facial action unit features as well as rigid and non-rigid facial shape parameters.

**Acoustic Modality**: COVAREP [7] is employed for the extraction of a series of acoustic features. These features encompass 12 Mel-frequency cepstral coefficients, pitch tracking, speech polarity, glottal closure instants, and spectral envelope, etc. Extracted from the full audio clip of each utterance, these features form a sequence that reflects the dynamic changes in vocal tone over the course of the utterance.

**Language Modality**: For the MSA task, following the state-of-the-art methods [57], DeBERTa [16] is employed to extract high-level textual representations. For the MHD and MSD tasks, following the state-of-the-art methods [13, 35, 36], ALBERT [25] is applied as the language learning network. For MHD and MSD, we concatenate the punchline and context token sequences to generate the final input: $U_l = C_l \oplus [SEP] \oplus P_l$, where the $[SEP]$ token is

---

[1]iMotions 2017. https://imotions.com/

used to separate the context tokens $C_l$ from the punchline tokens $P_l$ [13].

For the CMU-MOSI dataset, the dimensionality of the language, acoustic, and visual features are 768, 74, and 47, respectively. For the CMU-MOSEI dataset, the dimensionality of the corresponding unimodal features are 768, 74, and 35, respectively. For the UR-FUNNY and MUStARD datasets, the dimensionality of features of the language, acoustic, visual and HCF modalities are 768, 60, 36, and 4 respectively. For the feature extraction of the HCF modality, please refer to [13] for more details.

## A.6 Experimental Details

We implement the proposed MOAC model using the PyTorch framework on an NVIDIA RTX2080Ti GPU with CUDA version 11.6 and PyTorch version 1.13.1. The training of MOAC is facilitated by the AdamW [31] optimizer. Please refer to Table 6 for detailed information on the hyperparameter settings employed in our experiments. We perform a comprehensive grid search with fifty random iterations to identify the optimal hyperparameters. We search for the the best batch size from {32, 40, 48, 50, 60, 64}, and we define the search spaces for learning rate and shared feature dimensionality $d$ as {1e-6, 2e-6, 3e-6, 4e-6, 5e-6, 1e-5, 2e-5} and {100, 150, 200, 256, 300, 512}, respectively. We choose the hidden dimensionality $d'$ from {100, 128, 150, 200, 256}, and we choose the dropout rate from {0.3, 0.4, 0.5, 0.6, 0.7, 0.8}. We choose $\gamma_1$ and $\gamma_2$ from {0.1, 0.2, 0.3, 0.4, 0.5, 0.6, 0.7, 0.8, 0.9, 1.0}. We define the search space for $\alpha$ and $\beta$ as {1e-4, 0.001, 0.005, 0.01, 0.05, 0.1}. The other hyperparameters are pre-defined. Notably, the number of sampled pairs for the label-level ordinal loss and feature difference loss is set to twice the batch size, which helps to improve the performance of the model.

The structures of the fusion network and the predictor are shown in Fig. 5.

## A.7 Baselines

We compare the proposed MOAC with the following competitive baselines:

(1) **Graph-MFN** [64]: It introduces a multimodal graph neural network designed to investigate interactions at the unimodal, bimodal, and trimodal levels;

(2) **Multimodal Factorization Model** (**MFM**) [51]: MFM factorizes multimodal features into two sets of independent factors, namely multimodal discriminative factors and modality-specific generative factors, which can learn meaningful multimodal representations and interpret factorized representations to understand the interactions that influence multimodal learning;

(3) **MultiModal InfoMax** (**MMIM**) [11]: MMIM concurrently optimizes the mutual information among diverse unimodal representations and between multimodal representations and individual unimodal representations, thereby enabling the acquisition of more informative multimodal representations;

(4) **Hybrid Contrastive Learning** (**HyCon**) [37]: HyCon incorporates intra-modal and inter-modal contrastive learning strategies to thoroughly investigate the interactions both within individual samples and across different samples/categories;

(5) **Unified MSA and ERC** (**UniMSE**) [19]: UniMSE unifies the tasks of multimodal sentiment analysis and multimodal emotion

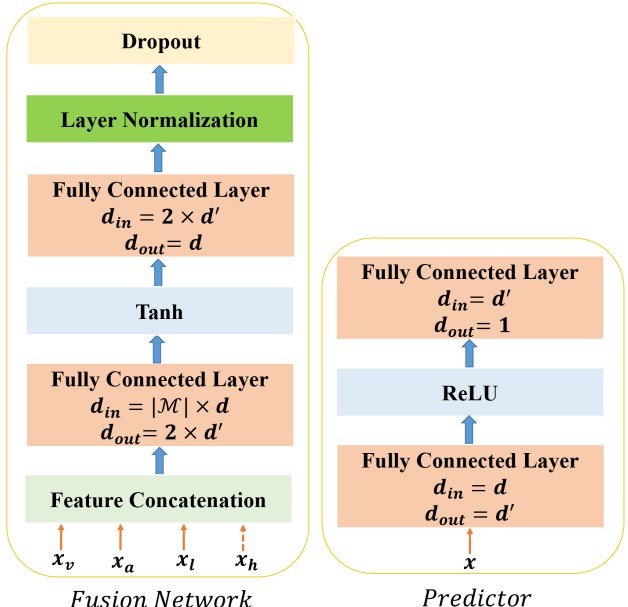

*Fusion Network*    *Predictor*

**Figure 5: The structures of the fusion network and the predictor.** $d'$ **represents the hidden dimensionality, and** $|\mathcal{M}|$ **denotes the number of modalities in the task with** $\mathcal{M}$ **being the modality set.**

recognition, casting them as generative tasks using the T5 [43] model;

(6) **Contrastive FEature DEcomposition** (**ConFEDE**) [60]: ConFEDE simultaneously performs contrastive representation learning and contrastive feature decomposition, thereby enriching the multimodal representation;

(7) **Trustworthy Multimodal Sentiment Ordinal Network** (**TMSON**) [58]: TMSON incorporates ordinal regression to establish a sentiment space that is aware of ordinal relationships, which is realized by a triplet loss;

(8) **Information-Theoretic Hierarchical Perception** (**ITHP**) [57]: Based on information bottleneck, ITHP designates a primary modality and treats the remaining modalities as detectors within the information pathway that serve to distill the flow of information;

(9) **Multimodal Adaptation Gate ALBERT** (**MAG-ALBERT**) [44]: MAG-ALBERT introduces a multimodal adaptation gate that enables large pre-trained transformer models to effectively process multimodal data during the fine-tuning phase;

(10) **Modality-Invariant and -Specific Representation** (**MISA**) [15]: MISA projects modality-specific and -invariant unimodal features into two distinct embedding subspaces for each individual modality;

(11) **Multimodal Global Contrastive Learning** (**MGCL**) [36]: MGCL conducts supervised contrastive learning based on multimodal representations and designs multiple operations to generate positive and negative samples for each multimodal representation;

(12) **Multimodal Correlation Learning** (**MCL**) [35]: MCL proposes supervised multimodal correlation learning task that is able

**Table 6: Hyperparameter Settings of MOAC. 'Dropout Rate' denotes the dropout rate of the dropout layer in the fusion network. The 'pre-training epochs' denotes the number of epochs for the first-stage training, and the 'training epochs' is the total number of epochs for the first and second stages.**

|  | CMU-MOSI | CMU-MOSEI | UR-FUNNY | MUStARD |
|---|---|---|---|---|
| Batch Size | 48 | 32 | 50 | 32 |
| Learning Rate | 1e-5 | 1e-5 | 4e-6 | 4e-6 |
| Training Epochs | 150 | 10 | 20 | 50 |
| Pre-training Epochs | 30 | 5 | 5 | 10 |
| $\lambda$ for Global Embedding | 0.95 | 0.95 | 0.95 | 0.95 |
| Shared Dimensionality $d$ | 128 | 128 | 300 | 256 |
| Hidden Dimensionality $d'$ | 150 | 100 | 128 | 128 |
| Dropout Rate | 0.5 | 0.3 | 0.6 | 0.6 |
| Margin $\gamma_1$ | 1 | 1 | 0.8 | 0.6 |
| Margin $\gamma_2$ | 0.1 | 0.1 | 0.1 | 0.1 |
| Regularization Loss Weight $\alpha$ | 0.005 | 0.005 | 1e-4 | 1e-4 |
| Feature-level Ordinal Loss Weight $\beta$ | 0.005 | 0.005 | 0.001 | 1e-4 |

to preserve the modality-specific information and learn more discriminative embedding space;

(13) **Humor Knowledge Enriched Transformer (HKT)** [13]: HKT is a promising method for MHD and MSD that incorporates humor centric feature to serve as external knowledge to deal with the ambiguity and sentiment presented in the language modality.

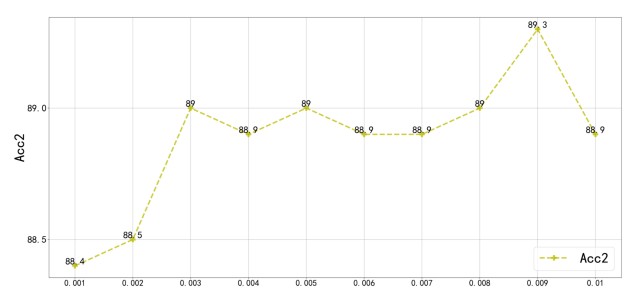

**Figure 6: Model performance w.r.t the change of $\alpha$.**

## A.8 Hyperparameter Robustness Analysis

In this section, we assess the impact of the hyperparameter $\alpha$, which denotes the weight of the regularization loss, on the CMU-MOSI dataset. The model's performance across various $\alpha$ values is depicted in Fig. 6. As shown in Fig. 6, MOAC delivers satisfactory performance when $\alpha$ is configured to a moderate value. Conversely, assigning a relatively small value to $\alpha$ results in a marginal decline in model performance. This is because when $\alpha$ is not sufficiently large, the contribution of the regularization loss is almost negligible, and the distributions of the predictive labels might deviate from the distributions of the annotated labels, which cannot provide a good parameter initialization for the model. Nevertheless, compared to baselines, MOAC generally achieves satisfactory performance in a wide range of hyperparameter settings for $\alpha$, which to some extent suggests the robustness of MOAC. Notably, when $\alpha$ is set to 0.009, the performance of the model exceeds the default setting (i.e., $\alpha = 0.005$). It demonstrates that the performance of

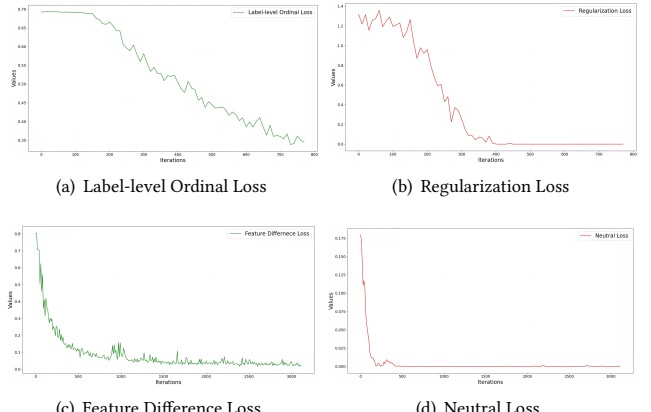

(a) Label-level Ordinal Loss (b) Regularization Loss

(c) Feature Difference Loss (d) Neutral Loss

**Figure 7: Learning curves of ordinal losses on CMU-MOSI.**

**Table 7: The testing results at the end of the first stage training on two datasets.**

| Dataset | Acc7 | Acc2 | F1 | MAE | Corr |
|---|---|---|---|---|---|
| CMU-MOSEI | 40.4 | 86.0 | 86.1 | 0.799 | 0.745 |
| CMU-MOSI | 39.0 | 87.3 | 87.3 | 0.746 | 0.806 |

the model can still be enhanced through further hyperparameter tuning, indicating the potential of MOAC.

## A.9 Learning Curves of Ordinal Losses on CMU-MOSI

In this section, we additionally provide the learning curves of ordinal losses on CMU-MOSI dataset in Fig. 7. In fact, the learning curves for the label-level ordinal loss, neutral loss, and regularization loss on the CMU-MOSI dataset are similar to those on the CMU-MOSEI dataset (see Section 4.5). While differently, the feature difference loss for the CMU-MOSI dataset approaches zero at the end of the second-stage training. This is due to the larger

number of training epochs in the second stage for CMU-MOSI, and the relatively smaller size of the CMU-MOSI dataset compared to CMU-MOSEI, which results in a higher degree of overfitting. This overfitting concern is the main reason we have chosen to display the learning curves on the CMU-MOSEI dataset in Section 4.5.

Moreover, we show the testing results at the end of the first stage training on both datasets in Table 7 as supportive results for the statement in Section 4.5. The results suggest that although we do not conduct the main task training in the first stage, the model still reaches satisfactory performance merely using the label-level ordinal learning, which further demonstrates the effectiveness of the proposed ordinal learning. Nevertheless, the relatively suboptimal results on Acc7 reveal that the label-level ordinal learning is coarse-grained, and that is why we need feature-level ordinal learning and main task learning to refine the predictions.

