# OpenReview forum: "Learning by Comparing: Boosting Multimodal Affective Computing through Ordinal Learning"
_ACM.org/TheWebConf/2025/Conference — WWW 2025 Poster_

### Official Review · Reviewer_rYvz · 2024-11-29

**Novelty:** 6
**Technical Quality:** 6

**Review:**

This paper proposes a Multimodal Ordinal Affective Computing framework, which include the ordinal information of affective concepts to learn more discriminative feature representations for three downstream tasks of MAC.

Pros:
1. The MOAC framework innovatively proposes neutral embedding, label-level ordinal learning, and feature-level ordinal learning methods to enhance multimodal affective computing by comparing.
2. The methodology and experiments sections are meticulously detailed, with a logical flow of language, figures, and tables that enhance clarity.

Cons:
1. I suggest the authors to introduce more related works about ordinal learning in multimodal affective computing, which help readers to known the latest researches.

[1] Xie Z, Yang Y, Wang J, et al. Trustworthy Multimodal Fusion for Sentiment Analysis in Ordinal Sentiment Space[J]. IEEE Transactions on Circuits and Systems for Video Technology, 2024.

2. Both the loss function of label-level ordinal learning and feature-level ordinal learning are based on fine-grained information of each sample. Why label-level is called coarse-grained learning, while feature-level is called fine-grained learning? What do you mean specifically by coarse and fine granularity? Is that an appropriate statement?

**Questions:**

1. In Figure 6, the performance of the model fluctuates with the change of parameter $\alpha$. What is the value range of $\alpha$? How does the model performance change when $\alpha$ is greater than 0.01?
2. MOAC is implemented on an NVIDIA RTX2080Ti GPU, What about the running time?

**Reviewer Confidence:**

3: The reviewer is confident but not certain that the evaluation is correct

**Scope:**

4: The work is relevant to the Web and to the track, and is of broad interest to the community

---

### Official Review · Reviewer_radr · 2024-11-29

**Novelty:** 5
**Technical Quality:** 5

**Review:**

This paper investigates the ordinal affect problem in multimodal affective computing by employing label-level ordinal learning to guide the model in initially perceiving the affective intensity of two samples. It further incorporates feature-level ordinal losses to encourage the model to learn ordered affective features, ultimately enhancing its performance in multimodal affective computing tasks.

Strengths:
1.The paper effectively designs two types of ordinal losses and regularization constraints, which ensure a deeper and finer-grained understanding of affective concepts.
2.It innovatively maintains and dynamically updates a neutral embedding. By calculating the difference between multimodal representations and the neutral embedding, the model achieves more precise affective predictions.
3.Beyond the experiments on Multimodal Sentiment Analysis (MSA), the paper also demonstrates the effectiveness of MOAC on Multimodal Humor Detection (MHD) and Multimodal Sarcasm Detection (MSD) tasks.

Weaknesses:
1.The proposed two-stage training approach is relatively cumbersome and involves numerous manually tuned hyperparameters, complicating the training and debugging process.
2.While the authors introduce a bias embedding and a neutral loss to refine the neutral embedding, the random selection of neutral samples raises concerns about the reliability and representativeness of the neutral embedding.

**Questions:**

1.In line 321, the paper mentions: |S𝑛| is the number of neutral samples in S𝑛 (S𝑛 is selected from a batch of data D). Could you elaborate on the specific batch selection strategy?

2.Have you considered leveraging more advanced PLMs? Do you anticipate performance improvements with their incorporation?

3.Paper [58] computes the ordinal relationships among triplet samples, which appears to be a more advanced approach for modeling ordinal structures. Could the authors clarify why this method was not adopted in the proposed framework? Were there specific challenges, limitations, or considerations that led to choosing the current approach instead?

**Reviewer Confidence:**

4: The reviewer is certain that the evaluation is correct and very familiar with the relevant literature

**Scope:**

3: The work is somewhat relevant to the Web and to the track, and is of narrow interest to a sub-community

---

### Official Review · Reviewer_JEqZ · 2024-12-02

**Novelty:** 4
**Technical Quality:** 5

**Review:**

This article proposes a Multi-Modal Ordinal Attribute Learning Framework (MOAC), focusing on the ordinal relationship characteristics in affective computing. The main contributions include:


1.Label-level ordinal learning: By comparing samples, the model's predictions align more closely with the ordinal nature of affective labels.


2.Feature-level ordinal learning: Through operations on feature differences and neutral embeddings, it refines and optimizes ordinal relationship features.


3.Superior performance on tasks like multi-modal sentiment analysis, humor detection, and sarcasm detection, demonstrating the framework's generality and effectiveness.


However, the writing lacks clarity and is often hard to follow, with subjective expressions that detract from readability. Particularly, the label-level prototype learning, which is claimed to simulate human reasoning, is difficult to comprehend and support.


Additionally, for the MHD and MSD tasks, the paper directly uses the approach from [1] to obtain "humor-related" information as a fourth modality. Here, MHD refers to humor detection and MSD to sarcasm detection. Using pre-existing methods for "humor-related" information in both tasks simultaneously is confusing. Furthermore, the paper does not explicitly provide experimental results comparing performance with and without this information.

In terms of experimental design, although the authors attempt to improve results through complex models and loss function designs, the overall performance improvement is limited, and lacks significance test (P-Value, Significance Level, etc.).


[1] Humor Knowledge Enriched Transformer for Understanding Multimodal Humor

**Questions:**

Please refer to the mentioned questions above.

**Reviewer Confidence:**

4: The reviewer is certain that the evaluation is correct and very familiar with the relevant literature

**Scope:**

3: The work is somewhat relevant to the Web and to the track, and is of narrow interest to a sub-community

---

### Official Review · Reviewer_Pnde · 2024-12-02

**Novelty:** 4
**Technical Quality:** 4

**Review:**

This paper leverages the ordinal nature of affective concepts for enhanced performance in multimodal affective computing tasks. The proposed methodology includes two main parts: The first is Label-level Ordinal Learning: Encourages the model to learn comparative relationships in the label space. The second is Feature-level Ordinal Learning: Introduces feature differences and a neutral embedding to capture ordinal relationships at the feature level.

**Questions:**

Strength:
- The paper is overall completed.
- The use of more loss for the MAC tasks.

Weakness:
- I do think they need more explanation about their motivation. The current version is likely to prove the motivation by citing [47,61] in the introduction. I think it is not intuitive, maybe adding a real example will be good.
- There exists some work that uses Ordinal Learning in Affective Computing. It seems that the author mentions some technique differences but does not say why these differences are necessary.
- The improvement of results is not very significant.

**Reviewer Confidence:**

2: The reviewer is willing to defend the evaluation, but it is likely that the reviewer did not understand parts of the paper

**Scope:**

2: The connection to the Web is incidental, e.g., use of Web data or API

---

### Official Review · Reviewer_pXzB · 2024-12-03

**Novelty:** 4
**Technical Quality:** 5

**Review:**

The paper addresses an important limitation in existing multimodal affective computing research, specifically the neglect of the ordinal nature of affective states. The authors propose the Multimodal Ordinal Affective Computing (MOAC) framework, which innovatively incorporates both label-level and feature-level ordinal learning to address this issue. The paper demonstrates a clear understanding of the problem and provides a comprehensive framework to enhance the modeling of ordinal relationships in affective computing tasks.

Strength:
1.The paper effectively highlights the gap in existing studies by emphasizing the importance of ordinal learning for affective states, which adds a unique and valuable perspective to the field.
2.The framework is validated on a regression task (multimodal sentiment analysis) and extended to classification tasks (multimodal humor and sarcasm detection), demonstrating strong generalizability across different problem domains.

 I’m not an expert in this area. Based on my experience, I think this is a complete research work with good quality.

**Questions:**

No Questions

**Reviewer Confidence:**

2: The reviewer is willing to defend the evaluation, but it is likely that the reviewer did not understand parts of the paper

**Scope:**

3: The work is somewhat relevant to the Web and to the track, and is of narrow interest to a sub-community